# On the Photostability of Cyanuric Acid and Its Candidature as a Prebiotic Nucleobase

**DOI:** 10.3390/molecules27041184

**Published:** 2022-02-10

**Authors:** Luis A. Ortiz-Rodríguez, Sean J. Hoehn, Carlos E. Crespo-Hernández

**Affiliations:** Department of Chemistry, Case Western Reserve University, Cleveland, OH 44106, USA; sxh905@case.edu

**Keywords:** cyanuric acid, prebiotic era, photostability, RNA ancestors, excited-state calculations

## Abstract

Cyanuric acid is a triazine derivative that has been identified from reactions performed under prebiotic conditions and has been proposed as a prospective precursor of ancestral RNA. For cyanuric acid to have played a key role during the prebiotic era, it would have needed to survive the harsh electromagnetic radiation conditions reaching the Earth’s surface during prebiotic times (≥200 nm). Therefore, the photostability of cyanuric acid would have been crucial for its accumulation during the prebiotic era. To evaluate the putative photostability of cyanuric acid in water, in this contribution, we employed density functional theory (DFT) and its time-dependent variant (TD-DFT) including implicit and explicit solvent effects. The calculations predict that cyanuric acid has an absorption maximum at ca. 160 nm (7.73 eV), with the lowest-energy absorption band extending to ca. 200 nm in an aqueous solution and exhibiting negligible absorption at longer wavelengths. Excitation of cyanuric acid at 160 nm or longer wavelengths leads to the population of S_5,6_ singlet states, which have ππ* character and large oscillator strengths (0.8). The population reaching the S_5,6_ states is expected to internally convert to the S_1,2_ states in an ultrafast time scale. The S_1,2_ states, which have nπ* character, are predicted to access a conical intersection with the ground state in a nearly barrierless fashion (ca. ≤ 0.13 eV), thus efficiently returning the population to the ground state. Furthermore, based on calculated spin–orbit coupling elements of ca. 6 to 8 cm^−1^, the calculations predict that intersystem crossing to the triplet manifold should play a minor role in the electronic relaxation of cyanuric acid. We have also calculated the vertical ionization energy of cyanuric acid at 8.2 eV, which predicts that direct one-photon ionization of cyanuric acid should occur at ca. 150 nm. Collectively, the quantum-chemical calculations predict that cyanuric acid would have been highly photostable under the solar radiation conditions reaching the Earth’s surface during the prebiotic era in an aqueous solution. Of relevance to the chemical origin of life and RNA-first theories, these observations lend support to the idea that cyanuric acid could have accumulated in large quantities during the prebiotic era and thus strengthens its candidature as a relevant prebiotic nucleobase.

## 1. Introduction

The origin and prebiotic ancestral lineage of ribonucleic acid (RNA) has been a mystery that has captivated scientists from all research backgrounds for many years. A principal focus of the RNA-world hypothesis surrounds the probiotic synthesis of RNA and its plausible precursors. Several triazine derivatives have been identified from reactions performed under prebiotic conditions as well as in meteorites, making their formation and existence in early Earth as precursors highly probable [1,2]. Specifically, melamine, ammeline, ammelide and cyanuric acid (CA) have been identified to spontaneously form from a simple gas mixture of CO, H_2_ and NH_3_ [3,4]. Recently, Jeilani et al. proposed free radical-mediated mechanisms that lead to the formation of the triazine derivatives, melamine, ammeline, ammelide and CA based on density functional theory calculations (DFT) that are reasonable for prebiotic scenarios [5]. Their results showed that CA has the shortest mechanism for formation and that the keto-tautomer should be the most stable [5].

CA is the triazine derivative of barbituric acid (BA), which has been proposed as a prebiotic ancestor of canonical RNA [6,7]. Interestingly, CA was found to spontaneously self-assemble to form a stable cyclic hexamer [8]. Additionally, CA has been identified to form supramolecular structures when hydrogen bonded to 2,4,6-triaminopyrimidine (TAP), which has been proposed as a promising prebiotic ancestor of the RNA nucleobases [6,7]. The standard-state free energy of CA incorporated into the higher ordered polymers was found to be −3.3 kcal/mol, which corresponds well with the standard free energy between canonical nucleobases within folded RNA structures [9,10]. Furthermore, Karunakaran et al. investigated polymer formation of TAP modified with a hexanoic acid tail [11]. While individually these monomers are achiral, upon polymerization the superhelical structures were identified through circular dichroism to have alternating domains of chirality within the same sample. However, when the CA derivative was provided as mainly a specific enantiomer for the polymer formation, homopolymer superhelical structures resulted, similar to what was observed in duplex DNA [11]. This result, along with the structural similarity of CA: TAP, CA: melamine, and the typical Watson–Crick base pairing observed in the canonical nucleobases, supports the idea that CA could be a plausible prebiotic ancestor of RNA. 

Importantly, however, for CA to have played an important role during the prebiotic era on Earth, it would have needed to survive the extreme conditions of those times to accumulate in significant amounts. Of particular relevance to this investigation, protection against UV radiation would have likely been an important selection criterion [12]. During the early Earth, the environmental conditions were much different than those present today, including the absence of an atmospheric ozone layer. However, atmospheric gases such as water and CO_2_ attenuated high energy radiation below 6.2 eV (200 nm) [13,14]. Therefore, even though it has been shown that CA can be formed under both prebiotic and astrophysical conditions, its photophysical and photochemical properties should be studied to predict its potential availability and participation in prebiotic chemistry. 

In this study, we investigated the photostability likelihood of CA using DFT and its time-dependent variant (TD-DFT). The earliest time that life forms first appeared on Earth was around 4 billion years ago [15,16,17] and a recent study estimated the pH to be 6.6 [18] around that time. Therefore, we focus solely on the tri-keto tautomer of CA in this investigation because the pK_a_ of CA is 6.9 [9]. It is predicted that CA has minimal absorption at longer wavelengths than 200 nm, in agreement with the incomplete experimental data available for CA [19]. Furthermore, the calculations predict that excitation of the optically bright S_5,6_(ππ*) states should lead to ultrafast internal conversion to the ground-state through ^1^nπ*/S_0_ conical intersections, which—taken together with the calculation of relatively small singlet-triplet spin–orbit couplings—suggest that a significant population of long-lived reactive triplet states is unlikely. 

## 2. Results and Discussion

### 2.1. Ground-State Structure and Absorption Spectrum of CA

Figure 1 shows the optimized ground-state structure of neutral tri-keto CA microsolvated with three explicit water molecules forming hydrogen bonds with the hydrogen and oxygen atoms at the X3LYP/C-PCM/cc-pVDZ level of theory in water. The X3LYP is a double-hybrid functional based on the Lee–Yang–Parr correlation functional designed to improve the accuracy of nonbonded interactions such as hydrogen bonds [20,21]. As shown in Figure 1, the bond lengths between nitrogen–carbon and carbon–oxygen are 1.38 and 1.23, respectively, regardless of the bond examined and not surprisingly since the molecule has D_3_h symmetry. 

Vertical excitation energies for the first six excited singlet states of microsolvated CA are reported in Table 1 at the TD-CAM-B3LYP/C-PCM/def2-TZVP//X3LYP/C-PCM/cc-pVDZ level of theory in water. In addition to the range-separated hybrid functional, CAM-B3LYP, [22] two other range-separated hybrid functionals, ωB97X [23] and LC-PBE, [24] and the hybrid functional, PBE0 [25], were evaluated to calculate the vertical excitation energies to compare their performance. The vertical excitation energies obtained with the range-separated hybrid functionals (i.e., CAM-B3LYP, ωB97X and LC-PBE) are within the mean standard deviation error (0.2–0.3 eV) of vertical excitation energies obtained with electronic-structure methods [26,27] and thus, from this point forward, and for simplicity, we focus in the main text on the results obtained with the CAM-B3LYP functional. The vertical excitation energies for the first six excited singlet states of microsolvated CA with ωB97X and LC-PBE functionals are reported in Appendix A, whereas the results obtained with the PBE0 functional are reported in Appendix A. 

As shown in Table 1, there are states with equal energy, character, and oscillator strength due to the high symmetry of the molecule (D_3_h). Based on the oscillator strengths of the transitions, only the S_5_(ππ*) and S_6_(ππ*) excited singlet states have a significant contribution to the absorption spectrum of CA in water. Hence, only these two states were considered in the calculated absorption spectrum of microsolvated CA reported in Figure 2. To generate the absorption spectrum, each transition was convoluted with a Gaussian function (FWHM = 16 nm). The absorption spectrum is a result of the linear combination of the individual Gaussian functions of each transition, which resulted in a Gaussian band with FWHM = 28 nm. Since both states have the same vertical energy and oscillator strength, the same result is obtained by multiplying the Gaussian function of one of the states by two. This calculated absorption spectrum agrees with the absorption tail reported by Sancier et al. for CA at pH 4.4 [19]. The use of Gaussian functions with FWHM < 16 nm to convolute the individual transitions was also attempted but did not accurately describe the absorption tail reported by Sancier et al. [19]. Furthermore, the FWHMs of the fully-resolved lowest-energy absorption band reported by Sancier et al. [19] for CA at pH 9.7 (i.e., for deprotonated CA) and that of the similar compound TAP [6] at pH 7.4 are ~25 nm. Therefore, we consider the methodology used herein to model the lowest-energy absorption band of neutral CA to be reasonable. Accordingly, as shown in Figure 2, CA has an absorption maximum at 160 nm and has a tail that extends to ca. 205 nm based on our modeling of the lowest-energy absorption band. Notably, the absorption coefficients above 200 nm are small (<1.5 × 10^3^ M^−1^ cm^−1^) and a comparatively small absorption is expected if CA is irradiated at longer wavelengths than 200 nm. 

### 2.2. Plausible Photochemical Deactivation Pathways of CA in Water

Vertical excitation energies were calculated for microsolvated CA at the TD-CAM-B3LYP/C-PCM/def2-TZVP//X3LYP/C-PCM/cc-pVDZ level of theory in water (Table 1) to estimate the absorption spectrum (vide supra) of CA, but also to characterize the ordering of the states in the Franck–Condon region and to propose plausible electronic transitions for intersystem crossing to the triplet manifold. Hence, the vertical excitation energies for excited triplet states that are isoenergetic to or lower in energy than the S_5_,_6_(ππ*) states are also reported in Table 1. We considered excited states equal or lower in energy than the S_5_,_6_(ππ*) state because, as shown in Figure 2, if CA were to absorb radiation around its absorption maximum of ca. 160 nm, the S_5,6_(ππ*) states would be the primary electronic states to be populated. Note that the S_4_ to S_1_ states are predicted to have negligible oscillator strengths (independent of their electronic character) according to the calculations, and, hence, should not be populated significantly upon direct absorption of electromagnetic radiation. We also remark that the character of the excited states reported in Table 1 is based on the inspection of the primary one-electron transition Kohn–Sham orbitals for each electronic state. 

Based on El-Sayed’s propensity rules [28,29], and assuming that the population reaching the S_5_,_6_(ππ*) states will internally convert to the S_1_,_2_(ππ*) states in an ultrafast time scale (i.e., obeying the Kasha’s rule) [30], the S_1,2_ → T_5_ and S_1,2_ → T_1,2_ electronic transitions could in principle compete with internal conversion from the S_1_,_2_(ππ*) states (i.e., by radiative and/or nonradiative processes) to the ground state. Hence, to further evaluate the probability of intersystem crossing pathways, the spin–orbit couplings between these transitions are reported in Table 2 for microsolvated CA at the TD-CAM-B3LYP/C-PCM/def2-TZVP//X3LYP/C-PCM/cc-pVDZ level of theory in water. We note that the magnitude of the spin–orbit coupling elements between other singlet–triplet excited state combinations presented in Table 1 with energy gaps lower than 0.5 eV are significantly smaller (ca. < 0.1 cm^−1^) than those shown in Table 2 (not shown). As displayed in Table 2, the magnitudes of the spin–orbit couplings are small (ca. 6 to 8 cm^−1^) but not necessarily negligible and may contribute to a very low yield of triplet state population. We note that the magnitude of the spin–orbit couplings predicted herein for CA are within the same order of magnitude as those reported for TAP (5 to 10 cm^−1^) [31], which has been shown by Brister et al. [6] to be highly photostable due to ultrafast fast relaxation pathways leading to a ground state in an aqueous solution. In addition, Rankine [31] located five different S_1_/S_0_ minimum-energy crossing points (MECP) for TAP in agreement with the ultrafast deactivation and photostability reported by Brister et al. [6]. Furthermore, the molecular structure of CA resembles that of uracil, but with a carbon atom replacing the N5 atom and the addition of a carbonyl at the C6 position. Uracil monomers are photostable due to the availability of accessible conical intersections between the excited singlet states and the ground state [32,33,34,35,36,37], and they populate the triplet state with less than ca. 1% yield in an aqueous solution [38]. Thus, considering the body of work reported for TAP and for the uracil monomers, we propose that triplet state population in CA should be very small (if not insignificant) in water. 

To further support this idea, we decided to investigate the availability of conical intersections between the S_1_ and S_0_ states in CA, which—together with the results reported above—should provide a more complete picture of the primary relaxation mechanism. Figure 3 shows the S_1_ minimum and the geometry of a conical intersection ((S_1_/S_0_)_CI_) that were optimized with the linear response implementation of TD-DFT (LR-TD-DFT). The (S_1_/S_0_)_CI_ exhibits C-puckering and significant out of plane displacement of the oxygen atom. Hereafter, we will label the puckering as C6-puckering for simplicity (i.e., the carbon atom that is puckered corresponds to the carbon atom that was labeled as C6 in Figure 1. However, due to the symmetry of CA, this C-puckering also applies to the C2 and C4 atoms. To obtain an estimation of a putative energy barrier to access the conical intersection, we performed optimizations along the O6C6N1C2 dihedral angle to capture the C6 puckering and the out of plane displacement of the oxygen atom going from the S_1_ minimum-geometry to the (S_1_/S_0_)_CI_. The potential energy profiles are reported in Figure 4. As shown in Figure 4, the calculations predict that the (S_1_/S_0_)_CI_ can be accessed in a nearly barrierless (0.03 eV) fashion, which suggests that this conical intersection could play a major role in the deactivation of CA. Therefore, these results, together with the spin–orbit couplings reported above, suggest that CA should be highly photostable following photoactivation. 

At this point, it is important to highlight that we are aware of the limitations of linear response (LR) TD-DFT in predicting the correct dimensionality of conical intersections between the ground and excited states due the single-reference nature of this method [39]. Therefore, to further strengthen the idea of the availability of a (S_1_/S_0_)_CI_, we decided to employ the spin–flip (SF) approach of TD-DFT (SF-TD-DFT) [40,41]. The SF approach has been used extensively to overcome the problem with the conical intersections between the ground and excited states [40,41,42]. In SF-TD-DFT, the idea is to use a different reference for the state of interest. For instance, in SF-TD-DFT, the singlet states are generated via excitation from a triplet reference state. Several studies have shown the correct dimensionality of the conical intersections obtained with SF-TD-DFT [40,43,44]. Thus, we assume that this methodology should provide a qualitatively correct picture of the system. Figure 5 shows the S_1_ minimum and the geometry of a conical intersection ((S_1_/S_0_)_CI_) that was optimized using SF-TD-DFT. As the (S_1_/S_0_)_CI_ found with LR-DT-DFT, this conical intersection exhibits C-puckering and significant out of plane displacement of the oxygen atom. However, the C-puckering and the displacement of the oxygen atom are more pronounced in the (S_1_/S_0_)_CI_ obtained using SF-TD-DFT. As shown in Figure 6, the (S_1_/S_0_)_CI_ can be accessed from the S_1_ minimum by overcoming a small energy barrier (0.13 eV). We note that this energy barrier is only 0.1 eV higher than the energy barrier found when LR-TD-DFT is used to find the (S_1_/S_0_)_CI._ Thus, the qualitative agreement between the LR-TD-DFT and SF-TD-DFT results, together with the vast body of work that has reported analogous conical intersections for similar molecules [32,33,34,35,36,37], strengthen the idea of the prospective availability of S_1_/S_0_ conical intersections and, as discussed above, the high photostability of this system. However, acknowledging the limitations of the methodology employed herein, we hope that these calculations will motivate theoretical groups that employ multireference methodologies to not only examine the availability of S_1_/S_0_ conical intersections such as the one reported here, but to also investigate whether the higher lying S_5_,_6_(ππ*) states have access to conical intersections with the S_1_ or the S_0_ state out of the Franck–Condon region. 

Finally, another photochemical pathway that should be considered is direct photoionization of CA, which can lead to the formation of reactive radicals and the eventual degradation of CA. To evaluate this possibility, we calculated the vertical ionization energy of microsolvated CA in water at the UX3LYP/C-PCM/cc-pVDZ level of theory. The predicted vertical ionization energy is equal to 8.2 eV (151 nm), or 791.2 kJ/mol (189.1 kcal/mol). Therefore, given that atmospheric gases such as water and CO_2_ attenuated high energy radiation at shorter wavelengths than 200 nm on the prebiotic Earth’s surface, the calculations predict that direct photoionization of CA should not have contributed to the degradation of CA in water. 

Figure 1 summarizes the proposed deactivation mechanism of CA in water, which is based on the calculations reported in this study. It is predicted that excitation at ca. 7.7 eV leads to the population of the S_5_,_6_ states. Subsequently, following the Kasha’s rule [30], the population internally converts to the S_1,2_ states since the possible intersystem crossing pathways from higher lying singlet states are not expected to play a major role in the deactivation of CA due to the negligible magnitude of their spin–orbit couplings. Following internal conversion from the high energy singlet states to the S_1,2_ states, internal conversion to the S_0_ state should occur, while intersystem crossing to the T_5_ or T_1,2_ states may participate to a very minor extent. Considering that the spin–orbit couplings of CA are of the same order of magnitude as those reported for TAP [31], which is photostable [6], and that a conical intersection between the S_1_ and S_0_ can be accessed in a nearly barrierless fashion, it is expected that internal conversion to the ground-state should play a major role in the deactivation of CA, while intersystem crossing to the triplet manifold should not. Collectively, the absorption spectrum shown in Figure 2, the vertical excitation energies reported in Table 1, the spin–orbit couplings reported in Table 2, and the potential energy surface profiles reported in Figure 4, provide compelling computational evidence for the high photostability of CA (tri-keto neutral form) in water under prebiotic conditions, and thus increase the probability of its prospective availability during the prebiotic era.

## 3. Computational Methods

All electronic-structure calculations were performed in ORCA 5.0.0 [45]. All optimizations were performed with the X3LYP functional [20]. The cc-pVDZ [46] basis set was used in all the optimizations. The conductor-like polarizable continuum model (C-PCM) [47] was used to implicitly model bulk solvation. To model solvent effects explicitly, three water molecules were added to form hydrogen bonds with the hydrogen and oxygen atoms. To increase the efficiency of the calculations, the RIJCOSX density fitting approximation was employed [48] This density fitting approximation decomposed the usual four-centered two-electron integrals into three-centered integrals using the corresponding auxiliary basis set. The energy of the radical cation to calculate the vertical ionization energy was obtained by performing a single-point calculation at the UX3LYP/C-PCM/cc-pVDZ level of theory in water, using a charge of +1 and a multiplicity of 2 on the optimized ground-state geometry. The vertical ionization energy was then obtained by subtracting the energy of the ground-state form that of the radical cation. 

Vertical excitation energies for relevant singlet and triplet states from the optimized ground state geometry were calculated using the CAM-B3LYP [22] functional and the def2-TZVP basis set [49]. Assignment of the respective character of the excited states was done by considering the oscillator strengths of the states and by visual inspection of the Kohn–Sham orbitals. Spin–orbit coupling matrix elements were computed using the Franck–Condon geometry at the TD-CAM-B3LYP/C-PCM/ def2-TZVP in water. The geometry of the lowest-energy singlet state was optimized at the TD-X3LYP/C-PCM/cc-pVDZ level of theory in water. The conical intersection between the lowest-energy singlet state and the ground state was optimized using LR-TD-DFT and SF-TD-DFT at the TD-X3LYP/C-PCM/cc-pVDZ in water. To estimate the energy barrier to access the conical intersection calculated with LR-TD-DFT, optimizations along the O6C6N1C2 dihedral angle going from the S_1_ minimum-geometry to the (S_1_/S_0_)_CI_ were performed. To estimate the energy barrier to access the conical intersection calculated with SF-TD-DFT, a minimum-energy path calculation between the S_1_ minimum-geometry to the (S_1_/S_0_)_CI_ was performed at the TD-X3LYP/C-PCM/cc-pVDZ in water using the Nudged Elastic Band method as implemented in ORCA 5.0.0 [50]. The choice to use the minimum energy path calculation over optimizations along the O6C6N1C2 dihedral angle to get the energy barrier to access the (S_1_/S_0_)_CI_ was made because this calculation was less computationally expensive for the SF-TD-DFT approach. All molecular structures reported in this work were visualized with Avogadro [51].

## 4. Conclusions

In this study, DFT and TD-DFT were used to evaluate the likelihood of the photostability of cyanuric acid in water. The calculations suggest that cyanuric acid should be photostable. Specifically, it is shown that the magnitude of the spin–orbit couplings between singlet and triplet states are small, which suggests that intersystem crossing should not play a major role in the deactivation of cyanuric acid. It is also shown that the lowest-energy excited singlet state of cyanuric acid can access a conical intersection with the ground state in a nearly barrierless (≤0.13 eV) fashion. Moreover, we show that photoionization of cyanuric acid is unlikely considering the high vertical ionization energy (8.2 eV) and the electromagnetic radiation that was available in the prebiotic era on the Earth’s surface. Therefore, these results provide compelling computational evidence that lends support to the idea that cyanuric acid could have accumulated in large quantities during the prebiotic era, which thus strengthens its candidature as a prebiotic nucleobase. 

## Data Availability

The data of this study is available from the corresponding authors, C.E.C.-H and L.A.O.-R, upon reasonable request.

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
