# Peer review of "On the Photostability of Cyanuric Acid and Its Candidature as a Prebiotic Nucleobase"

_molecules, 2022, doi:10.3390/molecules27041184_

Round 1

Reviewer 1 Report

The authors report a computational study on the Photostability of Cyanuric Acid, employed DFT. 

There are several issues which need to be considered :

  1. It is known that any photophysical processes should be computationally studied by employing multireference methods. During the process the GS (ground state) is contaminated and thus Single Reference methods (like DFT) fail to provide correct picture. For example, Figure 4, as it is presented is wrong, what is the Horinzontal axis (any reaction coordinate ?) The " straight"  line in Fig 4 is strange ! Conical Intersections can not be predicted by DFT, unless the authors provide clear references.
  2. Why the authors select only 3 water molecules for the explicit model? Under which criteria ? 
  3. The methodology is incomplete ? Which code and/or software was used for the computations (e.g for SOC ?). This prevents any reproducibility of the data.

Revise appropriately the manuscript and make clear the concept. The computational prorocol followed to produce any data is not the right one. 

The authors should try to support their following statement" We also hope that these calculations will motivate theoretical 
groups that employ multireference methodologies...." 

Author Response

Reviewer 1

Comment 1.

It is known that any photophysical processes should be computationally studied by employing multireference methods. During the process the GS (ground state) is contaminated and thus Single Reference methods (like DFT) fail to provide correct picture. For example, Figure 4, as it is presented is wrong, what is the Horinzontal axis (any reaction coordinate?) The " straight" line in Fig 4 is strange! Conical Intersections cannot be predicted by DFT, unless the authors provide clear references.

Reply #1: We thank the reviewer for the comment; however, we respectfully disagree. We understand that single-reference methodologies have their limitations. However, saying that any photophysical process should be computationally studied by a multireference method is not accurate. Important physical insights can still be obtained using single-reference methods. In the past, we (and many others) have successfully implemented TDDFT to complement our experimental results and subsequently, our computational results have been validated with multireference methodologies performed by other groups.  Additionally, we are careful to mention the limitations of the methodology employed in the manuscript.

Having broadly reply to the reviewer’s statement, however, and to further improve the results that are reported in the manuscript, we decided to also find the conical intersection using Spin-Flip TD-DFT. Several studies have shown the correct dimensionality of the conical intersections obtained with SF approach (Chem. Phys. Lett., 2001, 338, 375–384, Phys. Chem. Chem. Phys., 2010, 12, 12811–12825., J. Phys. Chem. A, 2009, 113, 12749–12753.) In the revised manuscript we added these references supporting the methodology. Furthermore, the horizontal axis in figure 4 is now labeled.

Comment 2.

Why the authors select only 3 water molecules for the explicit model? Under which criteria ?

Reply #2:

Previous work on similar molecules has used 2 to 4 water molecules to explicitly model solvation (Phys. Chem. Chem. Phys., 2017, 19, 17531, J. Am. Chem. Soc. 2006, 128, 2, 607–619) and provided accurate results. It is virtually impossible to know with certainty how many water molecules are needed to model the so-called first solvation shell. This has been broadly debated in the computational literature. In our case, we chose 3 water molecules because with 3 water molecules you can form hydrogen bonds with all six hydrogen bonding sites of cyanuric acid (neutral tri-keto).

Comment 3.

  1. The methodology is incomplete ? Which code and/or software was used for the computations (e.g for SOC ?). This prevents any reproducibility of the data. Revise appropriately the manuscript and make clear the concept. The computational prorocol followed to produce any data is not the
    right one. The authors should try to support their following statement" We also hope that these calculations will motivate theoretical groups that employ multireference methodologies...."

Reply #3:

We thank the reviewer for the comment, but we respectfully disagree. Our methodology is thorough and detailed.  As stated in the original manuscript, all calculations were performed in ORCA 5.0.0. We do not understand what the reviewer means by supporting the statement about motivating other theoretical groups to run multireference calculations. We think that it is clear that performing further calculations using multireference theories is not only ideal but necessary.

Reviewer 2 Report

The discussion of manuscript can be classified as abiogenesis topic. The significance is obvious. Cyanuric acid plays an important role in RNA-world hypothesis. And the plausible formation pathways of cyanuric acid have been reported intensively. The authors discussed the survival likelihood of cyanuric acid under UV radiation condition. This manuscript has a clear conclusion that cyanuric acid could be photostable. The conclusion is supported by evidence from TD-DFT calculation. TD-DFT can calculate excited state properties of regular organic compound accurately. Meanwhile, the authors also compared the calculation results with experimental data. I think the evidence is sufficient to support the conclusion. This is a high-quality manuscript. The figures are clear and pretty. And I have a smooth reading experience during the reviewing.

I have several comments and suggestions about this work. One suggestion is about the model. Only consider cyanuric acid and 3 waters cluster may be insufficient. For explicit solvent environment, including the first solvent shell is a better choice. Current computational power enables the larger scale TD-DFT calculation. For the discussion of photostability, nonadiabatic molecular dynamics simulation is used widely. I believe your work still has room for improvement and attracting more interest. And I also have several minor reversion suggestions.

Suggestion 1:

Figure 1,3: Molecular structures

Most molecular visualization software has citation requisition, although it’s not mandatory. Could you please cite their software if needed?

Suggestion 2:

Section 3 (Computational methods). Parts of computational details (functional, basis set) have already been introduced in the section 2. And the parts are discussed again in section 3. The sentence, “The X3LYP is an extended functional based on ……”, appeared twice (page 3 line93 and page 9 line 264).

Meanwhile, the statement, “X3LYP is an extended functional”, is uncommon. We usually call X3LYP as a double hybrid functional.

Author Response

Reviewer 2.

 We thank the reviewer for noticing that our work is of high-quality.

Comment 1:

Why 3 water molecules?

Reply #1:

Previous work on similar molecules have used 2 to 4 water molecules to explicitly model solvation (Phys. Chem. Chem. Phys., 2017, 19, 17531, J. Am. Chem. Soc. 2006, 128, 2, 607–619) and provided accurate results. Please, see also reply #2 to reviewer 1.

Comment 2:

Figure 1,3: Molecular structures Most molecular visualization software has citation requisition, although it’s not mandatory. Could you please cite their software if needed?

Reply #2: We thank the reviewer for the comment. We are now citing the software used.

Comment 3:

Section 3 (Computational methods). Parts of computational details (functional, basis set) have already been introduced in the section 2. And the parts are discussed again in section 3. The sentence,  “The X3LYP is an extended functional based on ……”, appeared twice (page 3 line93 and page 9 line 264). Meanwhile, the statement, “X3LYP is an extended functional”, is uncommon. We usually call X3LYP as a double hybrid functional.

Reply #3. We thank the reviewer for the comment. We have fixed this in the manuscript.

Reviewer 3 Report

I consider the scientific contribution of the authors as quite high. However. this is one concern that I would like to share.  According to results published in Phys. Chem. Chem. Phys., 2011,13, 4311-4317 the population of bifurcated hydrogen bonds during the hydration of species that are similar to those considered in this manuscript is rather low.  In the water solution, they hydrated using a single hydrogen bond rather than a bifurcated one. Therfore I would suggest to the authors to make some estimates how such change  in geometry of the hydration  will affect the obtained results.    

Author Response

Reviewer 3:

I consider the scientific contribution of the authors as quite high. However. this is one concern that I would like to share. According to results published in Phys. Chem. Chem. Phys., 2011,13, 4311- 4317 the population of bifurcated hydrogen bonds during the hydration of species that are similar to those considered in this manuscript is rather low. In the water solution, they hydrated using a single hydrogen bond rather than a bifurcated one. Therefore, I would suggest to the authors to make some estimates how such change in geometry of the hydration will affect the obtained results.

Reply: We thank the reviewer for the comment. Previous work on similar molecules have used 2 to 4 water molecules to explicitly model solvation as we did in our work (Phys. Chem. Chem. Phys., 2017, 19, 17531, J. Am. Chem. Soc. 2006, 128, 2, 607–619) and provided accurate results. Therefore, we do not think that it is necessary to perform additional calculations. Furthermore, the article that the reviewer mentioned (Phys. Chem. Chem. Phys., 2011,13, 4311- 4317) solely focuses on the impact of bifurcated or non-bifurcated hydrogen bonding in regards to proton transfer and tautomerization and their associated rate, neither of which are focus of our paper. Furthermore, all the optimizations reported in this work allowed the unconstrained optimization of all nuclear coordinates (including the water molecules) but the dihedral angle that was systematically changed to obtain the energy barrier to access the conical intersection.  Therefore, we do not think that the presence of singly bonded water molecules or bifurcated hydrogen bonds between explicit water molecules and the CA chromophore should have a major impact on the optimized geometries reported.

Round 2

Reviewer 1 Report

The authors have revised and made clear several theoretical issues raised in their work. However, it is known that DFT has serious drawbacks on the study of the excited states and should be used with caution. A thorough study on the adequacy of the functional should be followed and comparison with more accurate methods is, sometimes, mandatory. DFT is not a "black box" for the excited states  and this is clearly revealed in the literature

See for example

J. Am. Chem. Soc. 2004, 126, 12, 4007–4016,

https://www.science.org/doi/10.1126/science.1158722,

Chemical Physics Letters 461 (2008) 338–342